# Effect of Structural Build-Up on Interlayer Bond Strength of 3D Printed Cement Mortars

**DOI:** 10.3390/ma14020236

**Published:** 2021-01-06

**Authors:** Tinghong Pan, Yaqing Jiang, Hui He, Yu Wang, Kangting Yin

**Affiliations:** College of Mechanics and Materials, Hohai University, Nanjing 211100, China; thpan@hhu.edu.cn (T.P.); wangyuhhu@hhu.edu.cn (Y.W.); ktying@hhu.cn.com (K.Y.)

**Keywords:** 3D printed mortars, interlayer bond strength, structural build-up, surface moisture, time gaps

## Abstract

Understanding the relationship between the intrinsic characteristics of materials (such as rheological properties and structural build-up) and printability and controlling intrinsic characteristics of materials through additives to achieve excellent printability is vital in digital concrete additive manufacturing. This paper aims at studying the effects of material’s structural build-up on the interlayer bond strength of 3DPC with different time gaps. Structural build-up can indirectly affect the interlayer bond strength by affecting the surface moisture of concrete. Based on the structural build-up of 3DPC, a new parameter, maximum operational time (MOT), is proposed, which can be considered as the limit of time gap to ensure high interlayer bond strength. Slump-retaining polycarboxylate superplasticizer (TS) slightly slows down the physical flocculation rate, but increases the maximum operational time of the cement paste. Nano clay significantly increases the sort-term structural build-up rate and has the function of internal curing and water retaining. Composite with nano-clay and TS can reduce the loss of surface moisture of 3D printed layers, prevent the formation of interface weak layer, and increase the interlayer bond strength between printed layers. This contribution can provide new insight into the design of 3D-printed ink with good extrudability, outstanding buildability, and excellent interlayer bond strength.

## 1. Introduction

In recent years, extrusion-based 3D printed cement mortars (3DPC) have attracted extensive attention and research, due to their many benefits to concrete construction, such as eliminating formwork, saving labor costs, reducing wastes and possessing high design-freedom [1,2]. Many requirements have been proposed to ensure the printability of 3DPC [3,4,5], including flowability (workability), extrudability, buildability and interlayer bond strength. These requirements often involve many contradicting features. One of the contradictions is the different rheological requirements before and after the cement mortar is extruded from the nozzle. The ideal 3DPC should have good fluidity and extrudability before deposition to ensure smooth extrusion and quickly harden after deposition to support the weight of itself and the upper layer [6,7]. In addition, there are also a contradiction between the buildability and the interlayer bonding strength. High kinetics of structural build-up is needed to bear the weight of the later printed layers, while a high structural build-up rate leads to a decrease of the interlayer bond strength [8]. Therefore, it is difficult to design 3D printing cement-based ink with good extrudability, outstanding buildability and excellent interlayer bond strength.

In the case of layer-by-layer printing in 3DPC, optimizing the kinetics of structural build-up is essential to ensure successful printing. Structural build-up is a kinetic process, in which the “strength” of fresh cement paste increases gradually with resting time due to the physical flocculation and chemical hydration reaction. Structural build-up rate has a great influence to the buildability. In the process of 3D printing, high structural build-up rate is an important guarantee for multi-layer accumulation [9]. However excessively high kinetics of structural build-up is not appropriate for the pumping and injection of concrete. If the structural build-up is too fast, the viscosity and yield stress increases and reaches a critical value quickly, any stoppage may result in blockage of pips [10]. Also, high kinetics of structural build-up is not appropriate during multi-layer casting of concrete, which would cause weak interfaces in the joint of upper and lower layers. This phenomenon is firstly observed in self compacting concrete (SCC) [11]. When a plastic “cover layer” is bonded to a plastic or quasi-plastic “substrate layer”, the structural build-up of both of them will significantly affect the interlayer bond properties [11,12]. This phenomenon is also observed in 3DPC [13,14].

In 3DPC, it is difficult eliminate the weak connections or defects between two adjacent layers, due to the special way of casting (printing layer by layer), which significantly affects the interlayer bond strength of 3D printed samples [15]. This is an important consideration and a significant limitation on the viability of concrete 3D printing for structural applications [8,16]. Recently, there are a number of variables have been proved to affect interlayer bond strength of 3DPC: namely, printing parameters [17,18], time gap between printed layers [5], surface moisture [19,20], effective contact area [21,22] and topological interlocking [23]. Nevertheless, those researches only take the effects of external environment on the interlayer bond strength of 3DPC into account, and scant information is available on the effect of the intrinsic characteristics of materials (such as rheological properties and structural build-up of materials) on interlayer bond strength of 3DPC. It is necessary to investigate the origins of the weak interfaces between the layers in 3DPC. Furthermore, according to the design requirements of 3DPC (pumpable, extrudable, buildable and excellent interlayer bond strength), establishing the relationship between the structural build-up and interlayer bond strength is needed, due to the structural build-up go through the whole concrete 3D printing process and has been proved to significantly affect the extrudability and buildability of 3DPC.

Therefore, this paper mainly studies the effects of structural build-up on the interlayer bond strength of 3D printed cement mortars. Slump-retaining polycarboxylate superplasticizer (TS) and polycarboxylate superplasticizers (PCE) are used to control the rheological properties and nano-clay is added to improve the thixotropy, the effects of which on both rheological parameters (viscosity and yield stress) and structural build-up of 3DPC are evaluated. In addition, the effect of structural build-up on the surface moisture and interlayer bond strength is evaluated. The work has given a new insight into the design of 3D-printed ink with good extrudability, outstanding buildability and excellent interlayer bond strength.

## 2. Materials and Methods

### 2.1. Raw Materials

Tap-water and ordinary Portland cement (OPC, Type II, 42.5 grade, Nanjing Conch Cement Co. Ltd., Nanjing, China) is used in this work. Attapulgite clay exfoliated into nanoparticles (called as nano clay, Nc) shaped with 135 nm average length and about 58 nm diameter is used. Nc is often used in cement-based materials as a thixotropic admixture [24,25]. The chemical compositions of OPC and Nc are given in Table 1. A commercially available manufactured sand (fineness modulus: 2.3, maximum aggregate size: 5 mm) is used as fine aggregate. The technical specifications of sand are listed in Table 2.

A self-synthesized slump-retaining polycarboxylate superplasticizer (TS) and a commercial polycarboxylate superplasticizers (PCE) were applied in this work. The commercial PCE is provided by the Sobute New Materials Co. Ltd (Nanjing, China). The TS is prepared via semi-batch free radical polymerization at 80 °C by co-polymerizing the monomer of acrylic acid (AA), sodium allyl sulfonate (SAS), Polyoxyvinyl unsaturated ester macro-monomer (MPEG400MA) and α-methallyl-ω-hydroxy poly (ethylene glycol) ether macro-monomer with Mw of ca. 2400 (HPEG2400). Mixing these monomers with a monomer molar ratio of 3 (AA): 1 (SAS): 0.3 (MPEG400MA): 0.7 (HPEG2400). Ammonium persulfate was used as initiator, and its dosage was 40% of the total mass of monomer. The infrared spectrogram of PCE polymer and TS polymer are shown in Figure 1. The gel permeation chromatograph (GPC) apparatus of PCE polymer and TS polymer are shown in Appendix A The weight-average molecular weight (Mw) of PCE polymer and TS polymer are 38,993 g/mol and 46,960 g/mol, respectively. The number-average molecular weight (Mn) of PCE polymer and TS polymer are 19,930 g/mol and 23,863 g/mol, respectively.

### 2.2. Specimen Preparation

Two types of superplasticizers (TS and PCE, Solid content is 40%) are used in this work, and the dosage is fixed as 3‰ of cementitious materials. Nano-clay is also used as thixotropic component in the mixture, and the dosage varies between 0‰, 7‰, 8‰, 9‰ of cementitious materials. All mixture has the same water as cement ratio of 0.32 and bone gel ratio of 1:1.5. The proportion of 3D printed cement mortars is listed in Table 3.

JJ-5 planetary cement mortar mixer is used to mixing materials. Firstly, dry batch, such as cement, fine aggregate and nano clay are mechanically mixed for one minute in the cement mortar mixer to form a homogenous mixture. Next, the water and superplasticizer are poured into the mixer and mechanically mixed with dry batch for 60s at a slow speed. Then, stopping for 30 s to scrape off the residual slurry on the wall of the mixer. Lastly, the cement paste is quickly mixed for 90 s. The whole mixing process lasted for 3 min, since water was mixed with dry batch. The mixture is immediately placed into a specific container for rheological properties test.

### 2.3. Rheological Properties

The BROOKFIELD RST-SST rheometer and the medium size rotors (with a diameter of 15 mm and a length of 30 mm) is used to measure the rheological and thixotropic properties of fresh cement paste. In the rheological test, three divided specimens of the same batch are measured and the average value is calculated. After completing a test procedure, a new sample needs to be prepared for the next test.

Two rheological protocols are proposed to quantitatively measure rheological parameters and static yield stress, respectively.

#### 2.3.1. Yield Stress and Plastic Viscosity

This protocol is used to measure the rheological properties (yield stress and viscosity) of fresh cement pastes, as shown in Figure 2a. The specimen is first pre-sheared for 120 s by applying a shear rate sweep from 0 to 100 1/s, which is mainly used to create a uniform environment with little test error. Then, after 10 s of rest, an increasing shear rate ramp from 0 to 100 1/s within 100 s is applied to produce the up-curve of the flow test. Finally, the shear rate decreases from 100 to 0 1/s within 100 s to obtain the down-curve of the flow test. The viscosity and yield stress of cement paste can be obtained by fitting the down-curve of the flow test with Bingham model [26], as shown in Figure 2b.

#### 2.3.2. Static Yield Stress Test Protocol

This protocol is used to monitor the development of static yield stress with resting time. After mixing, mixture was divided into 10 groups to measure the value of static yield stress at different resting time. 10 groups of samples are put into different glass bottles, and then each group is manually stirred for 1 min to ensure that all samples have the same initial structure. After resting for a certain time (0 s, 150 s, 300 s, 450 s, 600 s, 750 s, 900 s, 1050 s, 2000 s, and 3000 s, respectively), the sample is taken out to test the static yield stress, as shown in Figure 3a. In the static yield stress test protocol, a constant shear rate of 0.2 1/s is used. The maximum shear stress on the shear stress vs time curve is defined as static yield stress [27], as shown in Figure 3b. According to the research results of Ivanova and Yuan et al. [28,29], the development of static yield stress can be used to monitor the structural build-up of cement-based materials. 

### 2.4. 3D Concrete Printing

A new type of meso-scale 3D printing concrete machine and A big size of the nozzle (5 cm in diameter) is used in this paper. Nozzle height (10 mm) and printing speed (15 mm/s) are kept constant in this work to ensure that structural build-up is the only dependent variable affecting interlayer bond strength of printed mortars. In order to avoid the influence of extrusion defects and component collapse deformation on the interlayer bond strength, Real 3D printing experiments were carried out to evaluate the printability of all of samples. All the samples mentioned in Section 2.2 were used in 3D printing test to evaluate the printability of materials, based on the actual printing status (whether the cement mortars can be continuously extruded and successfully stacked in 10 layers). The results showed that only three groups of mixtures (PCE+8‰Nc, TS+7‰Nc and TS+8‰Nc) met the printing requirements, as shown in Figure 4. Only these three groups of mixtures are considered in the subsequent study on the surface moisture of printed layer and interlayer bond strength between printed layers.

### 2.5. Surface Moisture of Printed Layer

JK-C10 Moisture meter is a digital moisture testing equipment, and mainly used for the surface humidity measurement of floor construction and other industries, such as gypsum boards, concrete, etc. In this paper, JK-C10 is used to measure the surface moisture of 3D printed layer at different resting time. At each resting time, three teat points are randomly selected to measure the surface moisture value, as shown in Figure 5, and an average value is calculated as the final value of surface moisture.

### 2.6. Tensile Bond Strength Measurement

Two-layers sample with a different time gap (2 min, 10 min, 20 min, 30 min and 40 min, respectively) is prepared for the measurement of interlayer bond strength. All the samples are printed using a 3-axis gantry printer. A square nozzle (40 × 20 mm opening) is used in this work. Nozzle standoff distance is maintained at a constant value of 20 mm and printing speed are kept at a constant value of 30 mm/s. After two-layer sample is printed, the strip sample is cut into small squares with a length of 40 mm, immediately. Then the cut specimens are cured in the standard curing environment for 3 days and 28 days respectively. Before the tensile loading test, the upper and lower ends of the specimen have been cut and polished, and the total thickness of the specimen after polishing is about 20 mm. Then, the samples are bonded to the molds with high-strength epoxy glue, as shown in Figure 6. In the tensile bond strength testing, tensile loading at a loading speed of 0.035 ± 0.015 MPa/s is applied to both the upper and lower ends of specimen. The tensile bond strength is calculated through the ratio of maximum tensile force to effective bonding area.

## 3. Results and Discussion

### 3.1. Rheological Properties

Yield stress and viscosity are two important parameters to characterize the rheological properties of fresh cement paste, which play a significant role in describe the workability of 3DPC and SCC [30]. The viscosity and yield stress of mixture with superplasticizer and nano-clay are obtained from the down-curve and are shown in Figure 7. In both samples with the addition of PCE and TS, the viscosity and yield stress increase significantly with the addition of nano clay. Nano clay has a fibrous structure, it distributes alternately and forms a disordered grid in the paste, which results in a rapid increase of viscosity and yield stress. In addition, nano clay will absorb the water in cement paste, thus reducing the volume fraction of the dispersed phase and increasing the viscosity and yield stress of the paste [31].

Furthermore, compared with the paste with PCE superplasticizer, the paste with TS has higher viscosity and yield stress. This shows that the water reducing and dispersing properties of TS is not as significant as that of PCE. PCE superplasticizer is often used to adjust the rheological properties and workability of fresh cement pastes. PCE has a long side chain, and its adsorption on the surface of cement particles often induces a steric hindrance effect, reducing the attractive interparticle forces between cement particles. Like PCE superplasticizer, TS superplasticizer also contains polyoxyethylene groups -(CH_2_CH_2_O)_m_-R as side chain. It can form a stable hydrogen bond with water to form a hydrophilic three-dimensional protective film, which can not only play the role of infiltration, but also provide steric hindrance. However, TS has no good coverage on the surface of cement particles due to its high molecular content, as shown in Appendix A. When TS is added into cement paste, a part of molecules intercalate into C_3_A layered hydration products to form calcium-aluminate-TS intercalation hydrate, which consumes and absorbs part of polymer molecules. Thus, the dispersion ability of TS is weaker than that of PCE, while the viscosity and yield stress of cement paste with TS are higher than that of mixture with PCE.

### 3.2. Evolution of Structural Build-Up

Static yield stress is an important rheological parameter and is indispensable in 3DPC. Its evolution with time is usually used to characterize the structural build-up of materials [29]. By testing the static yield stress of cement paste at different resting times, the structural build-up can be monitored [32,33], as shown in Figure 8.

The increase of the static yield stress with time can be described by two successive steps: (i) just after the mixture of cement with water, the static yield stress increases quickly until this increase slows down, followed by (ii) second static yield stress increases slowly with time. This evolution of the static yield stress can be fitted with a thixotropy model, as shown in Equation (1), which takes into account the different steps with five fitted parameters. Figure 9 shows the experimental results for static yield stress evolution with the model fitting and Table 4 lists the fitted parameters.
(1)$$\tau \left(t\right)=\tau_{0}+\tau_{1}\left(t\right)+\tau_{2}\left(t\right)=\tau_{0}+c\left(1+\left(\lambda_{flocs\_residual}-1\right)e^{-t/tr}\right)+A_{thix}t$$
where *τ*_0_ is initial static yield stress, *c* is a fitted parameter, the structural parameter *λ_flocs_residual_* is the degree of flocculation, which varies between 0 (fully broken-down state) 1 (fully interconnected state) [34], *tr* is the relaxation time, and *A_thix_* is the long-term structural build-up rate.

As long as mixing of cement and water, a rapid physical flocculation occurs due to the particle’s mobility and the attractive interactions between particles [35]. After mixing, there is less connection between cement particles and the connection force is weak. Then, the average number of contacts between cement particles increases quickly, due to the Brownian movement of particles, which results in a rapidly increase of static yield stress [36]. Once particles connect with each other and form large flocculent structures, the growth rate of static yield stress begins to decrease, due to the newly formed clusters being less mobile than single particles [37]. This step can be described by Equation (2).
(2)$$\underset{t\ll t_{perc}}{\lim}\tau \left(t\right)=\tau_{0}+c\left(1+\left(\lambda_{flocs\_residual}-1\right)e^{-t/tr}\right)$$

In this test, as soon as the mixture is taken out from the mixer, the evolution of structural build-up is monitored. Therefore, it is assumed that the cement past is fully broken-down, i.e., *λ_flocs_residual_* = 0. Thus, Equation (2) can be simplified as.
(3)$$\underset{t\ll t_{perc}}{\lim}\tau \left(t\right)=\tau_{0}+c\left(1-e^{-t/tr}\right)\approx R_{thix}t$$
where *R_thix_* is the approximate linear growth rate of short-term structural build-up. There is a turning point (*t_perc_*) between two dynamic processes, in which the short-term structural build-up rate is equal to the long-term structural build-up rate, as shown in Figure 9 and Equation (4).
(4)$$\{\begin{array}{c} \frac{d\tau_{1}\left(t\right)}{dt}=\frac{dc\left(1+\left(\lambda_{flocs\_residual}-1\right)e^{-t/tr}\right)}{dt}=\frac{c\left(1-\lambda_{flocs\_residual}\right)}{tr}e^{-t/tr}\\ \frac{d\tau_{2}\left(t\right)}{dt}=\frac{dA_{thix}t}{dt}=A_{thix}\\ \frac{d\tau_{1}\left(t\right)}{dt}=\frac{d\tau_{2}\left(t\right)}{dt}\Rightarrow \frac{c\left(1-\lambda_{flocs\_residual}\right)}{tr}e^{-t/tr}=A_{thix} \end{array}$$
where, *c* is a fitted parameter, the structural parameter *λ_flocs_residual_* represents the degree of flocculation in cement paste in this case, *tr* is the relaxation time, and *A_thix_* is the long-term structural build-up rate.

Solving Equation (4), the expression of *t_perc_* value can be obtained:(5)$$t_{perc}=-tr\times \ln\left(\frac{A_{thix}\times tr}{c\left(1-\lambda_{flocs_{residual}}\right)}\right)$$

In addition, the approximate linear growth rate of short-term structural build-up *R_thix_* can be calculated as follows:(6)$$R_{thix}=\frac{\tau_{t_{perc}}-\tau_{0}}{t_{perc}}$$

Furthermore, when cement particles touch with water, dissolution of the cement and precipitation of the hydrates on the surface of the particles occur. The precipitation of the hydrates increases the total surface of particles and the bond force between particles, which leads to a linear increase of the static yield stress. When the physical flocculation process gradually faded, the precipitation of the hydrates gradually dominated. This step can be approximated by the Equation (7),
(7)$$\underset{t\gg t_{perc}}{\lim}\tau \left(t\right)=\tau_{1}+A_{thix}t$$
where *τ*_1_ is different from the initial static yield stress used in Equations (1) and (2), which is no specific physical mean, and *A_thix_* is the long-term structural build-up rate.

The short-term structural build-up is controlled by the flocculation process of cement particles and its origin is purely physical. Superplasticizers can be used to control the flocculation rate of cement particles [38]. Superplasticizers can be adsorbed to the surface of the solid particles, which hinders flocculation of cement suspensions, due to the electrostatic repulsion and steric hindrance. From Table 4, it can be found that the initial static yield stress of mixture with TS polymer is higher than that of mixture with PCE polymer. This phenomenon described above is due to the fact that TS polymer has lower adsorption capacity and dispersion effect than PCE polymer. As long as TS polymer is added into the cement paste, polymer molecules intercalates into the C_3_A hydration layered products, forming calcium aluminate-superplasticizer intercalation hydrates, which consumes a lot of TS polymer molecules. Furthermore, it can also be observed from Table 4 that the mixture with TS polymer possesses higher value of *t_perc_* than the mixture with PCE polymer, while the *R_thix_* value of mixture with TS polymer is little lower than that of mixture with PCE polymer. Once PCE polymer is added into cement paste, all the polymer molecules are exposed, which will be consumed quickly due to the deposition of hydration products covers the polymer molecules adsorbed on the surface of cement particles. This results in a great loss of workability and high physical flocculation rate. Unlike PCE polymer, TS polymer molecules are gradually released into cement suspension system, due to the TS polymer molecules stored in calcium aluminate-superplasticizer intercalation hydrates being released gradually and the eater functions of TS polymer are hydrolyzed with cement hydration. This is why TS polymers prolong the physical flocculation process.

Furthermore, nano clay can significantly increase physical flocculation rate, which is of great significance for improving the buildability of 3DPC and reducing the formwork pressure of SCC [24,39]. Nano clays can absorb water that would normally help lubricate the microstructure, which increase the volume fraction of solid phase and increases the static yield stress of mixture. Nano clay fills the gap between cement particles or paste agglomerates, which increase the number of physical contact points and would lead to higher structural build-up rate. Therefore, adding nano clay into cement paste significantly increases the short-term structural build-up rate, and the more nano clay is added, the faster the short-term structural build-up is, as shown in Figure 8.

The long-term structural build-up has been attributed to the early hydrates formed between cement grains, developing a network connected by C-S-H bridges [34,36]. Compared with PCE polymer, TS polymer decreases the structuration rate of mixture, due to the higher retardation for cement hydration. Higher carboxyl density of TS polymer consumes large amounts of Ca^2+^ ions through forming complexes, which reduces the concentration of Ca^2+^ ions in cement paste and delays the formation of hydration products such as Ca(OH)_2_ and C-S-H. Furthermore, the addition of nano clay causes a high structuration rate. Thus, it can be speculated that nano clay has an immediate effect on short-term structural build-up and a relatively moderate effect on long-term structural build-up.

In all the experiments the values of *t_perc_* which represents the transition from physical flocculation to chemical structuration is in the range of 600 s to 1000 s. We define it as the maximum operational time (MOT) of 3DPC. When the resting time is less than MOT, the reversible physical flocculation process is dominant, and the cement mortar has good pumpability. In this case, the MOT has the same physical meaning as the open time proposed by T.T. Le, et al. [40], both of them are determined as the time period in which the workability of fresh mortar is at a level maintaining extrudability. When the resting time exceeds MOT, the irreversible chemical structural process is dominant, and the mortar loses plasticity gradually. Mortars mixed with TS copolymer have higher MOT than using PCE copolymer for delaying of chemical structural process and prolonging of physical flocculation process by slow-releasing of polymer molecules. So, it can be explained why TS polymers have better slump retention performance. Though nano clay can significantly promote the flocculation of cement particles, it also causes a rapid loss of fluidity and shortens the MOT of 3DPC.

### 3.3. Interlayer Bond Strength

The interlayer bond strength of mixtures with different additives and printing time gap are shown in Figure 10. It can be seen that the interlayer bond strength decreases slowly at first, then followed with a rapidly decreasing with the increasing of printing time gap. The initial interlayer bond strength (i.e., the interlayer bond strength under continuous printing condition) is the highest, 1.92 MPa for TS+8‰Nc, 1.86 MPa for TS+7‰Nc and 1.88 MPa for PCE+8‰Nc respectively at 28 days. When the printing time gap is less than 10 min, the decrease of interlayer bond strength is not remarkable, except for the mixture PCE+8‰Nc. At 28 days, the interlayer bond strength of PCE+8‰Nc sample with time gap of 10 min (1.72 MPa) is 8.5% lower than the initial interlayer bond strength (1.88 MPa). When the time gap is higher than 20 min, the interlayer bond strength decreases rapidly. At 28 days, the interlayer bond strengths of TS+8‰Nc samples with time gap of 30 min (0.94 MPa) and 40 min (0.67 MPa) are respectively 51.0% and 65.1% lower than the initial interlayer bond strength (1.92 MPa), the interlayer bond strengths of TS+7‰Nc samples with time gap of 30 min (0.88 MPa) and 40 min (0.64 MPa) are respectively 52.7% and 65.6% lower than the initial interlayer bond strength (1.86 MPa), the interlayer bond strengths of PCE+8‰Nc samples with time gap of 30 min (0.77 MPa) and 40 min (0.55 MPa) are respectively 59.0% and 70.7% lower than the initial interlayer bond strength (1.88 MPa). At three days, the change of interlayer bond strength is similar to that at 28 days.

The initial rheological properties, structural build-up and surface moisture of fresh cement mortars are considered to be the indispensable factors affecting the interlayer bond strength [11,12,19]. As soon as the 3DPC is extruded from the nozzle, significant moisture can be seen on the surface of the deposition layer, because the extrusion process brings extra moisture to the surface of the deposition layer [41]. In the extrusion process, the shearing of nozzle wall leads to the migration of particles and the formation of lubrication layer at the interface between the 3DPC and the wall of the nozzle, which leads to more water migration to the surface of the deposition layer, as shown in Figure 10. The rheological properties of suspension significant influence the migration of particles and the formation of lubrication layer [42]. The cement paste with PCE polymer has a lower viscosity and yield stress than that with TS polymer, which is beneficial for the migration of particles and the formation of lubrication layer during extrusion. Thus, the cement paste with PCE has more water on the surface of deposition layer than that with TS polymer. If the water accumulated on the surface of the deposition layer can’t evaporation rapidly, it will increase the water to cement ratio of 3DPC interface film layer, and reduce the interlayer bond strength. Thus, when the time gap is less than 10 min, the interlayer bond strength of 3DPC with PCE polymer is lower than that with TS polymer, due to more water accumulating on the surface of the deposition layer.

However, the surface moisture of the deposition layer decreases continuously with the increase of resting time, due to the evaporation of the surface moisture and the hydration of cement. As shown in Figure 11, the decrease of the surface moisture of the deposition layer with resting time also can be described by two successive steps. Firstly, the physical flocculation process is dominant, and the loss of surface moisture mainly depends on the water evaporation. In this work, all printing operations are carried out in a closed and non-ventilated environment, with a relatively slow water evaporation rate. Therefore, the surface moisture of deposition layer decreases slowly. When the resting time exceeds MOT, the precipitation of the hydrates gradually dominated, and some early hydration products, such as ettringite and C-S-H gel are generated, which consumes the water in the cement paste. Thus, when the time gap beyond the MOT, the surface moisture decreases quickly due to high evaporation rate and cement hydration, which results in a rapid decrease in the interlayer bond strength of 3DPC. Therefore, we can infer that the structural build-up of the cement paste is an important factor affecting the surface moisture of the 3D printed layer. MOT can not only represent the transition from physical flocculation to chemical structure of cement paste, but also be used as the starting point of rapid decrease of surface moisture of printing layer. This can be used to explain why the surface moisture of cement paste with TS polymer decreases more slowly than that of PCE paste. Furthermore, under the standard curing conditions, the nano clay in 3DPC has functions of internal curing and water retention, which promotes the hydration of cement and raises moisture on the surface of the deposited layer. This can explain why the surface moisture of sample TS+8‰Nc is higher than that of sample TS+7‰Nc.

With the change of the surface moisture of the printed layer, the interlayer bonding strength of 3DPC also changes significantly, as shown in the Figure 10. There is a good correlation between the interlayer bonding strength and the surface moisture of the deposition layer, which is consistent with the research results of [19]. Compared with the cement paste with PCE polymer, the cement paste with TS polymer has a higher MOT value and better surface moisture retention ability, so its inter-layer bonding strength decreases slowly than that of the sample with PCE polymer.

In conclusion, there is a technic index link to the structural build-up and the interlayer bond strength of 3DPC. The MOT can be used as the upper limit of the time gap of 3DPC. Beyond the MOT, the cement paste will lose the activity of chemical bonding between upper and lower layers. The MOT proposed in this paper is similar to the open time proposed by other researchers, but there are also differences. The open time was determined as the time period in which the workability of fresh concrete was at a level that maintains extrudability [40], while the MOT can not only represent the period when the workability of fresh concrete is maintained at the level of compressibility, but also the time period when 3DPC multi-layer printed structure has good interlayer bond strength. Therefore, it is of great significance to select suitable additives, adjust the structural build-up rate and the MOT of 3DPC to design a new 3DPC with good extrudability, outstanding buildability, and excellent interlayer bond strength.

## 4. Conclusions

In this work, the interlayer bond strength of 3DPC is mainly investigated with respect to the structural build-up and initial rheological properties. The work has given a new insight into the design of 3D-printed ink with good extrudability, outstanding buildability and excellent interlayer bond strength. Based on the presented results, the following results have been obtained:Structural build-up of 3D printed cement mortars (3DPC) is an essential parameter of fresh cement mortars, and plays a key role to control the surface moisture of 3D printed layer and the interlayer bond strength of 3DPC.Based on the structural build-up of 3DPC, a new parameter (called maximum operational time, MOT) is proposed. When the time gap is less than MOT, 3DPC has good workability and high surface moisture content, which is conducive to the diffusion and adhesion between the upper and lower printed layer. When the time gap is higher than MOT, the interlayer bond strength decreases rapidly with the increase of the time gap, due to the rapidly loss of surface moisture of 3D printed layer. MOT is the limit of time gap to ensure high interlayer bond strength.Nano-clay, even a very small amount, will significantly increase structural build-up rate (both short-term structural build-up and long-term structural build-up), also decease the MOT of 3D printed mortars, which is associated with the thixotropic property of clay particles, responsible for rapid flocculation and agglomeration of particles.Slump-retaining polycarboxylate superplasticizer (TS) has lower dispersibility than polycarboxylate superplasticizers (PCE) at the initial stage, but it significantly increases the MOT of 3DPC. The improved performance is associated with the slow release effect of TS polymer, responsible for the better persistence of workability and plasticity.Composite using of nano-clay and TS can control the rheological properties, structural build-up rate and MOT, which can be used to produce 3D-printed ink with good extrudability, outstanding buildability and excellent interlayer bond strength.The addition of water retaining materials (such as nano-clay) is beneficial to the interlayer bond strength, due to the function of internal curing and water retaining, which promotes the hydration of cement and eliminates part of the shrinkage on the surface of the deposited layer, thus improving the microstructure of interface film layer and increasing the interlayer bond strength.

In future work, the effect of rheological properties and structural build-up on the surface moisture of concrete, heterogeneous hydration in interface film layer, and the micro-structure of interface film layer will be studied to obtain new insights into the interlayer bond strength of 3DPC.

## Figures and Tables

**Figure 1 materials-14-00236-f001:**
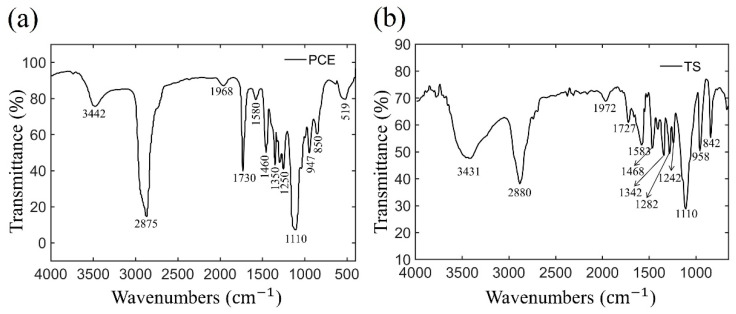
IR spectra for (**a**) polycarboxylate superplasticizers and (**b**) slump-retaining polycarboxylate superplasticizer.

**Figure 2 materials-14-00236-f002:**
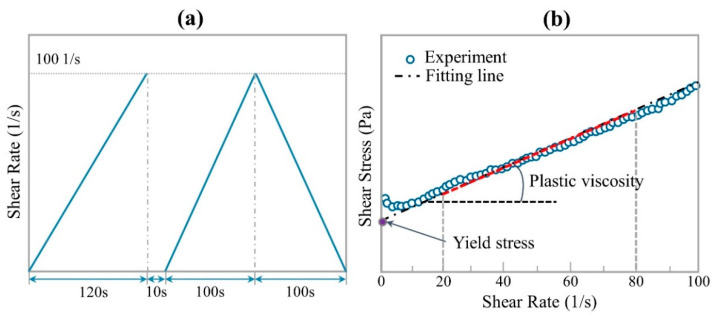
(**a**) Test procedure of rheological parameters; (**b**) the yield stress and plastic viscosity obtained by the Bingham model.

**Figure 3 materials-14-00236-f003:**
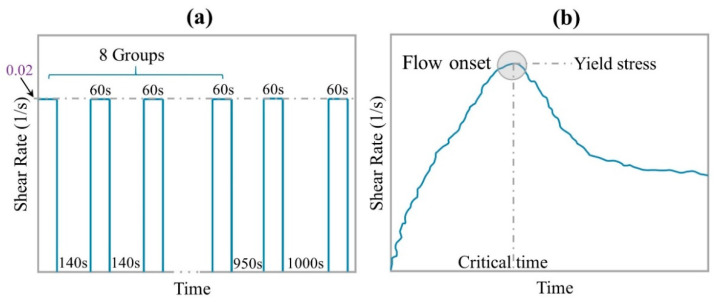
(**a**) Test procedure of static yield stress, and (**b**) static yield stress obtained from the peak value.

**Figure 4 materials-14-00236-f004:**
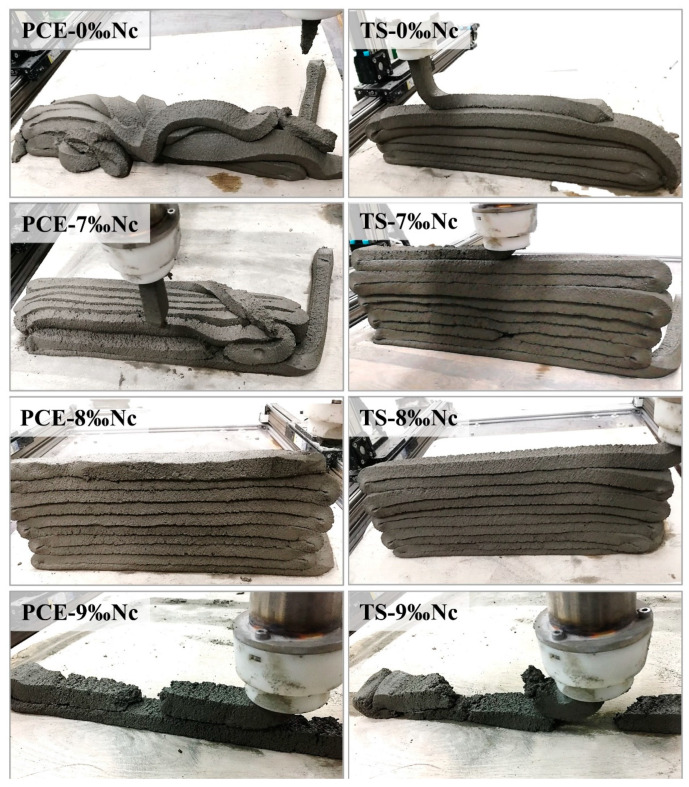
Demonstration of 3D printability of cement paste with different type of superplasticizers and different dosage of nano clay.

**Figure 5 materials-14-00236-f005:**
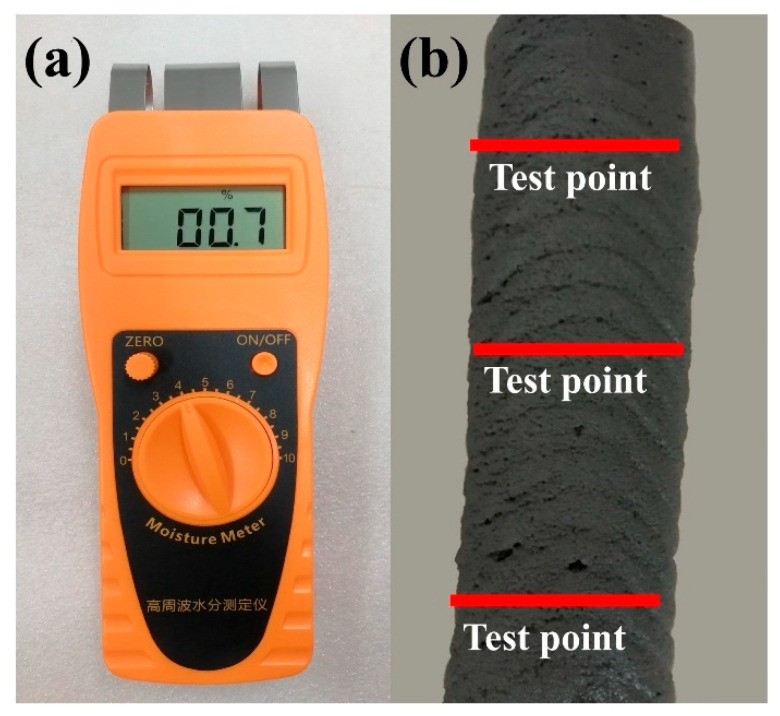
Surface moisture of 3D printed layer. (**a**) JK-C10 Moisture meter; (**b**) randomly selected three test points to measure the surface moisture value.

**Figure 6 materials-14-00236-f006:**
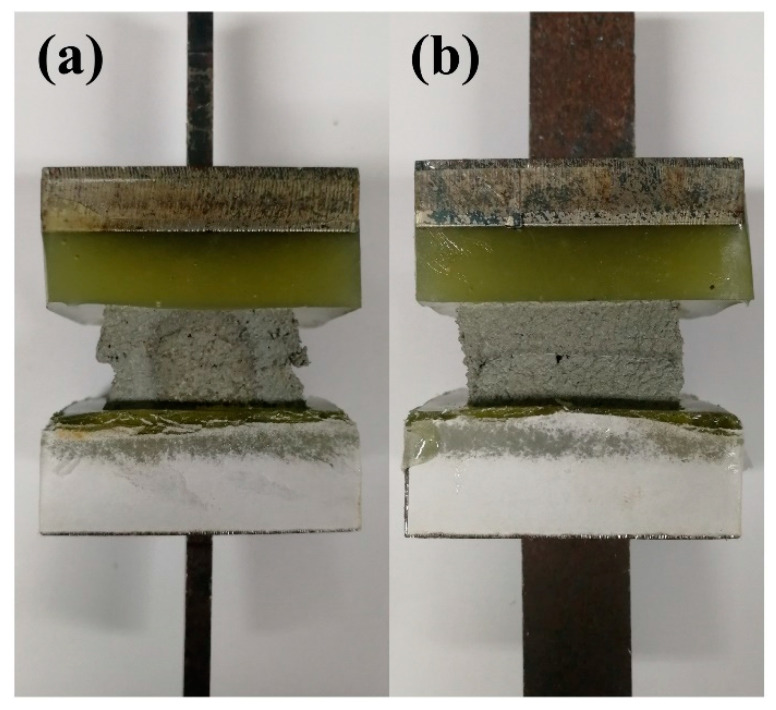
Schematic of sample preparation for tensile bond strength. (**a**) is the front view of the sample and (**b**) is the side view of the sample.

**Figure 7 materials-14-00236-f007:**
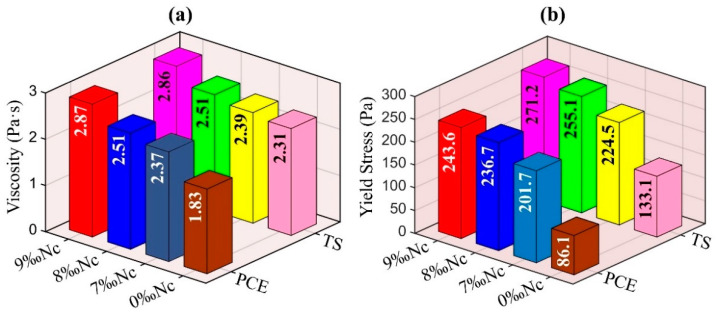
Rheological parameters (**a**) Viscosity and (**b**) Yield stress.

**Figure 8 materials-14-00236-f008:**
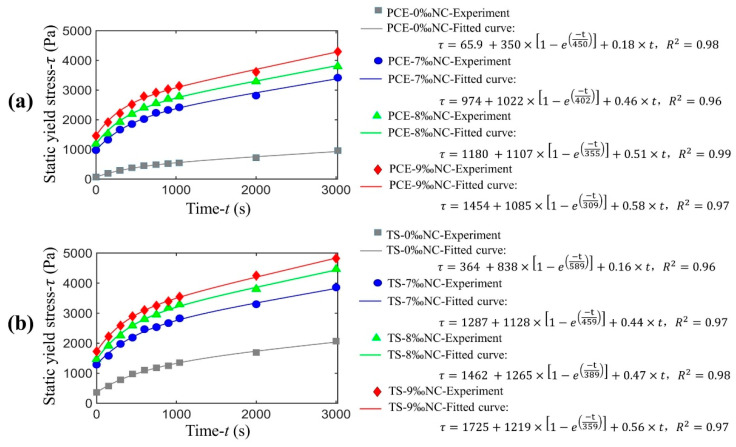
Experimental results and fitted curve of static yield stress of (**a**) mixture with PCE polymer, and (**b**) mixture with TS polymer with resting time.

**Figure 9 materials-14-00236-f009:**
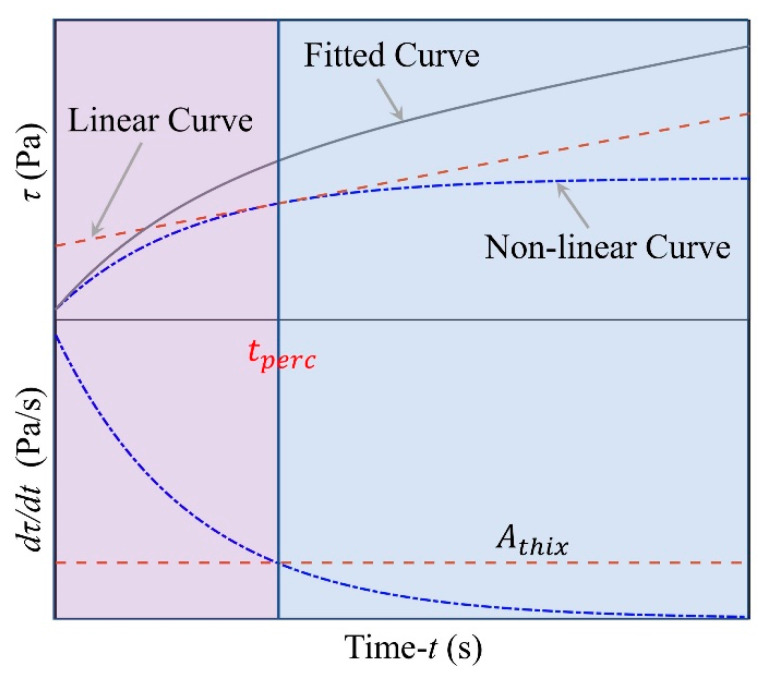
Schematic diagram for the calculation of *t_perc_*.

**Figure 10 materials-14-00236-f010:**
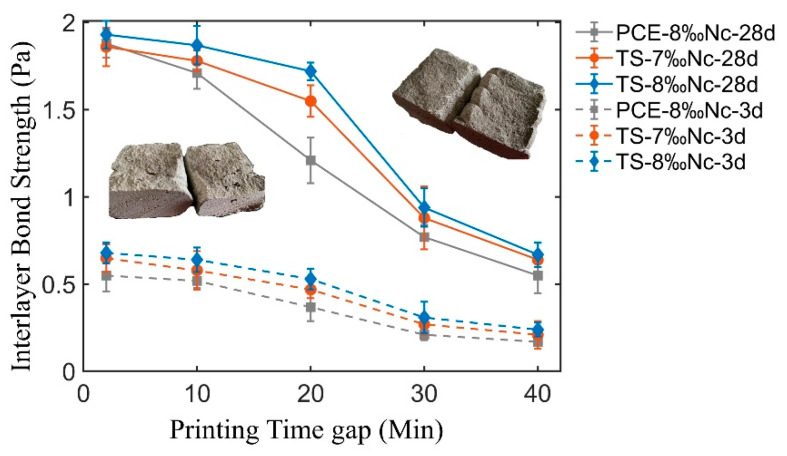
Interlayer bond strength vs. printing time gap of 3D printed mortar with standard cured for 3 days and 28 days.

**Figure 11 materials-14-00236-f011:**
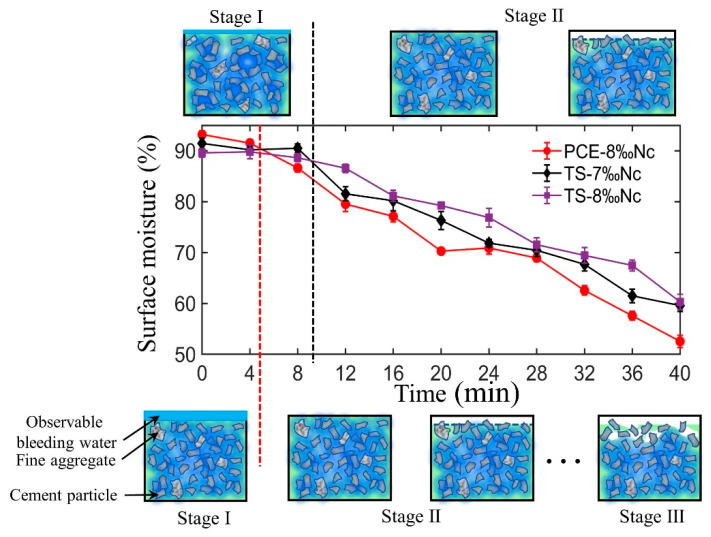
Surface moisture vs. resting time of 3D printed mortar. The schematic diagram of the surface moisture change of the mixture with TS polymer (above) and PCE polymer (below).

**Table 1 materials-14-00236-t001:** Chemical compositions of OPC (Type II) and Nc [wt.%].

Materials	CaO	SiO_2_	Al_2_O_3_	FeO_3_	Na_2_O	MgO	K_2_O	SO_3_	TiO_2_	L.O.I
OPC(Type II)	62.60	21.65	5.56	4.32	0.24	0.84	0.76	2.85	-	1.27
Nc	9.62	58.4	26.73	0.51	0.21	0.20	3.05	-	0.15	1.13

**Table 2 materials-14-00236-t002:** Technical specifications of sand.

Density (g/cm^3^)	Packing Density (g/cm^3^)	Water Absorption/%	Fineness Modulus	Maximum Particle Size/mm
OPC(Type II)	62.60	21.65	5.56	4.32
Nc	9.62	58.4	26.73	0.51

**Table 3 materials-14-00236-t003:** Proportion of 3D printed mortars.

Material	Cement	Quartz Sand	Water	Nano-Clay	One of TS or PCE Polymer
Quantity(g)	1400	1750	460	0; 9.8; 11.2; 12.6	4.2

**Table 4 materials-14-00236-t004:** Fitted parameters for each cement mixture.

No.	*τ*_0_ (Pa)	*R_thix_* (Pa/s)	*t_perc_* (s)	*A_thix_* (Pa/s)
PCE-0‰Nc	65.9	0.64	765.3	0.18
PCE-7‰Nc	974.5	1.87	687.3	0.46
PCE-8‰Nc	1180.1	2.34	642.78	0.51
PCE-9‰Nc	1454.2	2.61	556.4	0.58
TS-0‰Nc	363.7	1.08	957.8	0.16
TS-7‰Nc	1286.7	1.83	759.2	0.44
TS-8‰Nc	1462.1	2.47	677.4	0.47
TS-9‰Nc	1725.1	2.58	604.7	0.56

## Data Availability

Data sharing not applicable.

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
