# Peer review of "Effect of Structural Build-Up on Interlayer Bond Strength of 3D Printed Cement Mortars"

_materials, 2021, doi:10.3390/ma14020236_

Round 1
Reviewer 1 Report
In order to improve the quality of the submitted manuscript, we request the following revisions.
- Minor typos, errors: needs correction.(ex) 281 line: Missing table number )
- It is necessary to move the equation solving process in the appendix to the text.
- Correlation coefficient needs to be indicated in the regression analysis equation
- It is recommended to change to "tr" instead of "θ" in the regression equation (because it is the meaning of time).
- In Figure 10, the line area needs to be modified to a constant thickness.
- In the the results and analysis of the bond strength, it is necessary to analyze again including the results of 0, 9% of the amount of nano clay (NC) added.
- Regarding “MOT”, briefly mentioned in lines 308~320, 384~393 of the main text, while in the conclusion, it was described at a level beyond the contents of the main text. It is necessary to describe in more detail the “MOT” in the main text.
Author Response
Dear Ms. Stephanie Bai and dear reviewers,
Re: Manuscript ID: materials-1044379
Type of manuscript: Article
Title: Effect of structural build-up on interlayer bond strength of 3D printed cement mortars
Authors: Tinghong Pan, Yaqing Jiang *, Hui He *, Yu Wang, Kangting Yin
Received: 3 December 2020
E-mails: thpan@hhu.edu.cn, yqjiang@hhu.edu.cn, coolhehe@hhu.edu.cn,
wangyuhhu@hhu.edu.cn, ktying@hhu.cn.com
Thank you for your letter and for the reviewers’ comments concerning our manuscript entitled “Effect of structural build-up on interlayer bond strength of 3D printed cement mortars” (Manuscript ID: materials-1044379). Those comments are all valuable and very helpful for revising and improving our paper. We have studied comments carefully and have made corrections which we hope meet with approval. Revised portions are marked in red in the paper. The main corrections and the responds to the reviewer’s comments are as following:
Reviewer #1
- Comment:
Minor typos, errors: needs correction. (ex) 281 line: Missing table number )
Response:
Thank you for giving so many detailed and profound suggestions. We are sorry to make this mistake. We changed it and please see the revised manuscript (page 11, line 282).
- Comment:
It is necessary to move the equation solving process in the appendix to the text.
Response: Thank you for giving so many detailed and profound suggestions. The equation solving process in the appendix has been moved to the text, please see the revised manuscript (page 10, line 255-264).
- Comment:
Correlation coefficient needs to be indicated in the regression analysis equation
Response: Thank you for giving so many detailed and profound suggestions. Correlation coefficient has been indicated in the regression analysis equation. (ex) Figure 8. Please see the revised manuscript (page 9, line 239).
- Comment:
It is recommended to change to "tr" instead of "θ" in the regression equation (because it is the meaning of time).
Response: Thank you for giving so many detailed and profound suggestions. We changed almost all "θ" in the regression equation to "tr". Please see the revised manuscript.
- Comment:
In Figure 10, the line area needs to be modified to a constant thickness.
Response: Thank you for giving so many detailed and profound suggestions. In Figure 10, the color area represents the error area of the test data, so the width of the line varies. To increase the readability of the image, Figure 10 has been changed into an error bar diagram. Please see the revised manuscript (page 13, line 395).
- Comment:
In the results and analysis of the bond strength, it is necessary to analyze again including the results of 0, 9% of the amount of nano clay (NC) added.
Response: Thank you for giving so many detailed and profound suggestions. The cement mortar with a nano-clay content of 0 and 9% does not have good extrudability and buildability, which cannot be successfully extrusion and stacking, as shown in Figure 4. Therefore, this paper did not study the interlayer bonding performance of cement mortar with a content of 0, 9% nano-clay.
- Comment:
Regarding “MOT”, briefly mentioned in lines 308~320, 384~393 of the main text, while in the conclusion, it was described at a level beyond the contents of the main text. It is necessary to describe in more detail the “MOT” in the main text.
Response: Thank you for giving so many detailed and profound suggestions.
MOT is a hub that links structural build-up, surface moisture and interlayer bond strength of 3DPC. MOT can not only represent the transition from physical flocculation to chemical structure of cement paste, but also be used as the starting point of rapid decrease of surface moisture of printing layer. Furthermore, the MOT can be used as the upper limit of the time gap of 3DPC. Beyond the MOT, the cement paste will lose the activity of chemical bonding between upper and lower layers.
We re-describe the “MOT” in the main text, please see the revised manuscript (lines 309~320, 358~394).

Reviewer 2 Report
The aim of the research, try to find a link between the structural build-up of the layers and the interlayer bonding is a recent approach and needs more in depth investigations. Also the addition of nano particles is new.
However, the results are not represented in a clear way and this confuses the reader. For example (all other comments are indicated within the paper), a clear overview and explanation of the relationship between moisture content and MOT should be included as a first way to improve the quality of the paper
Author Response
Dear Ms. Stephanie Bai and dear reviewers,
Re: Manuscript ID: materials-1044379
Type of manuscript: Article
Title: Effect of structural build-up on interlayer bond strength of 3D printed cement mortars
Authors: Tinghong Pan, Yaqing Jiang *, Hui He *, Yu Wang, Kangting Yin
Received: 3 December 2020
E-mails: thpan@hhu.edu.cn, yqjiang@hhu.edu.cn, coolhehe@hhu.edu.cn,
wangyuhhu@hhu.edu.cn, ktying@hhu.cn.com
Thank you for your letter and for the reviewers’ comments concerning our manuscript entitled “Effect of structural build-up on interlayer bond strength of 3D printed cement mortars” (Manuscript ID: materials-1044379). Those comments are all valuable and very helpful for revising and improving our paper. We have studied comments carefully and have made corrections which we hope meet with approval. Revised portions are marked in red in the paper. The main corrections and the responds to the reviewer’s comments are as following:
- Comment:
The aim of the research, try to find a link between the structural build-up of the layers and the interlayer bonding is a recent approach and needs more in depth investigations. Also the addition of nano particles is new.
However, the results are not represented in a clear way and this confuses the reader. For example (all other comments are indicated within the paper), a clear overview and explanation of the relationship between moisture content and MOT should be included as a first way to improve the quality of the paper.
Response: Thank you for giving so many detailed and profound suggestions.
MOT is a hub that links structural build-up, surface moisture and interlayer bond strength of 3DPC. MOT can not only represent the transition from physical flocculation to chemical structure of cement paste, but also be used as the starting point of rapid decrease of surface moisture of printing layer. When the resting time is lower than MOT, the physical flocculation process is dominant, and the loss of surface moisture mainly depends on the water evaporation. When the resting time exceeds MOT, the precipitation of the hydrates gradually dominated, and some early hydration products, such as ettringite and C-S-H gel are generated, which consumes the water in the cement paste. Thus, when the resting time beyond the MOT, the surface moisture decreases quickly due to high evaporation rate and cement hydration.
We added part of the description to explain the relationship between moisture content and MOT, please see the revised manuscript (page 13, lines 358~377).

Reviewer 3 Report
Comment for the authors
This paper presents the effect of the hydration rate of different material on the bond strength and the interbonding strength of 3D printed concrete. There is a major issue and several minor changes before it is suitable for publication.
Major issue
1. Bleeding to the top surface of the material occurs when formwork is used. In the formwork, water usually appear at the top because it does not have any other direction to escape. It occurs when free water in the mix is pushed upward to the surface due to the settlement of heavier solid particles. However, for a printed filament, bleeding, if any, will only result water escaping from the sides of the filament. The reasoning in section 3.3 does not seems to be valid.
Minor issue
1. Cold joint is a term used to describe a plane of weakness in concrete when the first batch of concrete has begun to set before the next batch is added, so that the two batches do not intermix. In the case for concrete printing, cold joint does not occur. Kindly rephrase and choose your word carefully.
2. Section 2.4.3; what type of pre-experiment were carried out to evaluate the printability of the samples? How were they evaluated?
3. Figure 6b seems to be obtained from existing literature, kindly give credit to author from whom you copied.
4. Line 256; the particles that flocculates together do not form aggregates. Kindly rephrase.
Author Response
Dear Ms. Stephanie Bai and dear reviewers,
Re: Manuscript ID: materials-1044379
Type of manuscript: Article
Title: Effect of structural build-up on interlayer bond strength of 3D printed cement mortars
Authors: Tinghong Pan, Yaqing Jiang *, Hui He *, Yu Wang, Kangting Yin
Received: 3 December 2020
E-mails: thpan@hhu.edu.cn, yqjiang@hhu.edu.cn, coolhehe@hhu.edu.cn,
wangyuhhu@hhu.edu.cn, ktying@hhu.cn.com
Thank you for your letter and for the reviewers’ comments concerning our manuscript entitled “Effect of structural build-up on interlayer bond strength of 3D printed cement mortars” (Manuscript ID: materials-1044379). Those comments are all valuable and very helpful for revising and improving our paper. We have studied comments carefully and have made corrections which we hope meet with approval. Revised portions are marked in red in the paper. The main corrections and the responds to the reviewer’s comments are as following:
- Comment:
Major issue: Bleeding to the top surface of the material occurs when formwork is used. In the formwork, water usually appear at the top because it does not have any other direction to escape. It occurs when free water in the mix is pushed upward to the surface due to the settlement of heavier solid particles. However, for a printed filament, bleeding, if any, will only result water escaping from the sides of the filament. The reasoning in section 3.3 does not seem to be valid.
Response: Thank you for giving so many detailed and profound suggestions.
We carefully studied the concept of “bleeding”, consulted many articles about “bleeding”, and agreed that bleeding to the top surface of the material does not occur in the case for concrete printing. We reanalysis the change mechanism of surface moisture of the 3D printed layers.
In the extrusion process, the shearing of nozzle wall leads to the migration of particles and the formation of lubrication layer at the interface between the 3DPC and the wall of the nozzle, which leads to more water migration to the surface of the deposition layer. Thus, as soon as the 3DPC is extruded from the nozzle, significant moisture can be seen on the surface of the deposition layer. Then, with the increase of resting time, the surface moisture of the deposition layer decreases continuously, due to the evaporation of the surface moisture and the hydration of cement. When the resting time is lower than MOT, the physical flocculation process is dominant, and the loss of surface moisture mainly depends on the water evaporation. When the resting time exceeds MOT, the precipitation of the hydrates gradually dominated, and some early hydration products, such as ettringite and C-S-H gel are generated, which consumes the water in the cement paste. Thus, when the resting time beyond the MOT, the surface moisture decreases quickly due to high evaporation rate and cement hydration.
Please see the revised manuscript (page 13, lines 358~377).
- Comment:
Cold joint is a term used to describe a plane of weakness in concrete when the first batch of concrete has begun to set before the next batch is added, so that the two batches do not intermix. In the case for concrete printing, cold joint does not occur. Kindly rephrase and choose your word carefully.
Response: Thank you for giving so many detailed and profound suggestions. We carefully studied the concept of “cold joint”, consulted many articles about “cold joint”, and agreed that cold joint does not occur in the case for concrete printing. We chose a new word “weak interfaces” carefully to describe the 3D printing mortar interlayer bonding, and substitute “weak interfaces” for almost all “cold joint” in the text, please see the revised manuscript.
- Comment:
Section 2.4.3; what type of pre-experiment were carried out to evaluate the printability of the samples? How were they evaluated?
Response: Thank you for giving so many detailed and profound suggestions. In order to more intuitively represent the printable performance of cement mortar, we added the actual printing renderings of each group of samples in the paper, as shown in Figure 4. Based on the actual printing status (whether the cement mortars can be continuously extruded and successfully stacked in 10 layers), the printable performance of the material is evaluated.
- Comment:
Figure 6b seems to be obtained from existing literature, kindly give credit to author from whom you copied.
Response: Thank you for giving so many detailed and profound suggestions. Figure 6 has been redrawn, please see the revised manuscript (page 7, line 185).
- Comment:
Line 256; the particles that flocculates together do not form aggregates. Kindly rephrase.
Response: Thank you for giving so many detailed and profound suggestions. After reviewing the literature and discussing, we decided to replace the sentence " Once particles connect with each other and form large aggregates " with" Once particles connect with each other and form large flocculent structures ". Please see the revised manuscript (page 9, line 249).

Reviewer 4 Report
As it was compactly stated in the title, the paper presents research on the effects of material's structural build-up on interlayer bond strength of 3D printed cement mortars.
However, the originality and novelty of research activity is not well exposed and/or highlighted.
Also, there are required some changes in the graphs presented in the paper.
In Figure 7 some of the descriptions (values on top of the bars) are completely unreadable.
In Figure 8, the descriptions of the curves are written with the use of the too-small font.
In Figure 9, the results for TS+7%Nc-28d and TS+7%Nc-3d, as well as for PCE+8%Nc-28d and PCE+8%Nc-3d are marked with the same type of line (there are no differences).
Author Response
Dear Ms. Stephanie Bai and dear reviewers,
Re: Manuscript ID: materials-1044379
Type of manuscript: Article
Title: Effect of structural build-up on interlayer bond strength of 3D printed cement mortars
Authors: Tinghong Pan, Yaqing Jiang *, Hui He *, Yu Wang, Kangting Yin
Received: 3 December 2020
E-mails: thpan@hhu.edu.cn, yqjiang@hhu.edu.cn, coolhehe@hhu.edu.cn,
wangyuhhu@hhu.edu.cn, ktying@hhu.cn.com
Thank you for your letter and for the reviewers’ comments concerning our manuscript entitled “Effect of structural build-up on interlayer bond strength of 3D printed cement mortars” (Manuscript ID: materials-1044379). Those comments are all valuable and very helpful for revising and improving our paper. We have studied comments carefully and have made corrections which we hope meet with approval. Revised portions are marked in red in the paper. The main corrections and the responds to the reviewer’s comments are as following:
- Comment:
In Figure 7 some of the descriptions (values on top of the bars) are completely unreadable.
Response: Thank you for giving so many detailed and profound suggestions. Figure 7 has been redrawn,Please see the revised manuscript (page 8, line 223).
- Comment:
In Figure 8, the descriptions of the curves are written with the use of the too-small font.
Response: Thank you for giving so many detailed and profound suggestions. Figure 8 has been redrawn,Please see the revised manuscript (page 9, line 240).
- Comment:
In Figure 9, the results for TS+7%Nc-28d and TS+7%Nc-3d, as well as for PCE+8%Nc-28d and PCE+8%Nc-3d are marked with the same type of line (there are no differences).
Response: Thank you for giving so many detailed and profound suggestions. Figure 9 has been redrawn,Please see the revised manuscript (page 12, line 339).

Round 2
Reviewer 1 Report
The items requested for correction were well reflected.